# Conflict, healthcare and professional perseverance: A qualitative study in a remote hospital in an Anglophone Region of Cameroon

**Juste Ongeh Niba**[1,2☯]*, **Stewart Ndutard Ngasa**[2,3☯], **Neh Chang**[2,4‡], **Eric Sanji**[2‡], **Anne-Marie Awa**[5], **Therence Nwana Dingana**[2], **Carlson Babila Sama**[2], **Leticia Tchouda**[2], **Mekolle Enongene Julius**[6]

1 Clinical Research Education Networking and Consultancy (CRENC), Douala, Cameroon, 2 Medical Research and Career Organization, Oxford, United Kingdom, 3 Health Education for England, North West School of Psychiatry, Liverpool, United Kingdom, 4 Faculty of Health Sciences, University of Buea, Buea, Cameroon, 5 Saint Joseph Catholic Hospital (SJCHC), Widikum, Cameroon, 6 Cameroon Baptist Convention Health Services, Bamenda, Cameroon

☯ These authors contributed equally to this work.
‡ These authors also contributed equally to this work.
* justeniba237@gmail.com

**Data Availability Statement:** All relevant data are within the paper and its Supporting Information files.

## Abstract

Armed conflicts are a major contributor to global disease burden owing to their deleterious effects on health and healthcare delivery. The Anglophone crisis in Cameroon is one of the ongoing conflicts in Sub-Saharan Africa and has led to massive displacement of healthcare workers (HCWs). However, some HCWs have stayed back and continued working. An understanding of their experiences, perspectives and professional perseverance is lacking. We designed a phenomenological study using Focused Group Discussions (FGDs) and in-depth interviews to: understand the experiences of 12 HCWs in a remote hospital in the North West region of Cameroon with armed groups; evaluate how it affects healthcare delivery from HCWs perspective and examine HCWs coping mechanisms during the conflict with a view of informing HCW protection policies in conflict zones. Results revealed that HCWs go through all forms of violence including threats, assaults and murders. Overall insecurity and shortage of health personnel were major barriers to healthcare delivery which contributed to underutilization of healthcare services. Participants observed an increase in complications due to malaria, malnutrition and a rise in maternal and infant mortality. The hospital management and Non-Governmental Organizations (NGOs) played an essential role in HCWs adaptation to the crisis. Nevertheless they unanimously advocated for a cease fire to end the conflict. In the meantime, passion for their job was the main motivating factor to stay at work.

**Funding:** The authors received no specific funding for this work.

**Competing interests:** The authors have declared that no competing interests exist.

## Introduction

Since the end of the Second World War, there have been a declining number of wars around the world. However, armed conflicts and violence are currently on the rise, with many conflicts today waged between non-state actors such as political militias, criminal and international terrorist groups [1]. In every continent, unresolved regional tensions, a breakdown in the rule of law, illicit economic gain and the scarcity of resources exacerbated by climate change, have become major drivers of civil wars and international conflict [1]. In 2016, the United Nations reported that more countries experienced violent conflict than at any point in the last three decades. At the same time, conflicts are becoming more fragmented [1].

The Anglophone crisis in Cameroon is one of such fragmented conflicts. The conflict is marked by territorial contestation between separatists and the government of the Republic of Cameroon over the North West (NW) and South West (SW) regions. The origin of the crisis dates back to 1961 where both English (Southern) and French speaking parts of Cameroon agreed on a federated state with equal partnership and preservation of their respective socio-cultural heritages [2]. In late 2016, a peaceful protest by English speaking teachers and lawyers for the preservation of the English education curriculum and law courts led to hostile retaliation from the armed forces and later escalated into a full blown war between secessionists and the government [2].

Reports suggest that the armed conflict has led to the displacement of over 1.3 million people of whom about 70,000 are refugees in neighbouring Nigeria, with over 3000 fatalities [3]. The United Nations estimated over 2.2 million people living in these regions need humanitarian aid and over 600,000 children are deprived of effective schooling [4]. Massive displacement of persons might have major effects on all sectors including education, economy, and healthcare. The impact on the health sector has been further exacerbated by attacks on healthcare professionals and patients together with destruction of health facilities [5]. Instances of people being shot in healthcare facilities have also been reported. Such occurrences may lead to a breakdown of healthcare systems in these regions especially in remote areas. Access to healthcare in many remote villages and some urban towns is limited. Hence the available facilities are put under additional strain due to high demand and limited resources [5].

Despite all the challenges on ground, many healthcare facilities have remained functional and healthcare professionals have stayed back providing services to the non-displaced population. An understanding of the experiences, perspectives and professional perseverance of HCWs in an ongoing complex conflict like that of the Anglophone crisis in Cameroon is lacking.

We thus sought to understand HCWs' perseverance amid crisis through three questions:

1. What are the experiences of HCWs working in conflict zones with brutality?

2. What are the consequences of this on healthcare delivery?

3. How do HCWs respond to these challenges while preserving professionalism at work?

The ultimate goal is to provide local authorities and stake holders with comprehensive evidence that will allow them make informed decisions concerning the welfare of HCWs in conflict settings.

## Methods

To answer our research questions, we conducted a qualitative study in a hospital setting using the phenomenological approach. Data were obtained from participants using a semi-structured interview guide.

## Setting

Data were collected from January to Mach 2022 from health personnel working at the Saint Joseph Catholic Health Center (SJCHC) under the Batibo administrative Health District in the North West region of Cameroon. This was the only functional hospital in the town which was now almost fully under the control of the separatists. The hospital has a workforce of 32 HCWs with 20 nurses (62.5%), 6 midwives (18.75%), 4 laboratory technicians (12.5%) and 2 medical doctors (6.25%).

Widikum is a border town between the NW and SW Regions. This makes it a confluence for socio-economic activities between the regions and thus an area of interest for both the separatists and the government of Cameroon. It has been severely affected by the crisis, recording some of the worst health outcomes in the entire region [6].

## Participants

Health care workers in the aforementioned health facility were recruited to participate in the study. To be eligible for the study, participants had to be: 1) a HCW with $\geq$ 2 years of working permanently at the study site; 2) able to voluntarily consent to the study. Using the purposive and snowball sampling methods [7], we recruited a total of 12 participants (Table 1).

## Ethical considerations

Ethical approval was obtained from the Institutional Review Board of the Bamenda Regional Hospital in the North West Region of Cameroon. Participants provided informed consent prior to commencement of interviews.

## Data collection

We collectively designed a semi-structured interview guide which we used to conduct FGDs and individual interviews (Table 2). The guide was developed by members of the research team (Research and Med Career) and it covered the following topics: experiences of HCW with armed groups; brutality and barriers to healthcare delivery; and professional perseverance of HCWs during conflict.

**Focused group discussions.** A total of two FGDs were conducted by the principal investigator (first author) who had a previous experience in qualitative research methods. Each FGD comprised of 5 participants and lasted 45 minutes.

**Individual interviews.** Two face to face interviews were administered by the principal investigator and each session lasted 20 minutes.

All interviews were conducted in English language and this was done only after written consent was obtained. Before each interview commenced, participants filled out a socio-demographic questionnaire which included age, sex, marital status and profession (Appendix 1). We stopped interviewing participants when no new themes emerged [8].

**Table 1. Socio-demographic characteristics of participants.**

| Demographics | N = 12 |
|---|---|
| **Age, years (Mean±SD)** | **33.7±6.7** |
| Gender (female, %) | 50 |
| Marital status (married, %) | 50 |
| Profession (%) | |
| • Nurse | 66.7 |
| • Doctor | 25 |
| • Laboratory technician | 8.3 |

**Table 2. Semi-structured interview guide for participants.**

| |
|---|
| **A. Experiences with brutality from any armed groups** |
| 1. Have you had a confrontation or encounter with any of the armed forces? |
| If yes, can you tell me more about the incident? |
| *Probe 1*: What in your opinion provoked the incident? |
| *Probe 2*: Did you sustain any physical injuries? |
| 2. Did you make any official report to your superior on the incident? If no why not? If yes what was the outcome of the report? |
| 3. Any specific incidents or stories of brutality experienced by other health workers that you would like to share with us anonymously? |
| 4. Do you think health workers are specifically targeted? If yes in your opinion are they such an important target to all armed groups? |
| **B. Brutality and barriers to health care delivery** |
| 1. Do you think the incident has affected the way you work? |
| *Probe 1*: How often do you come to the hospital? |
| *Probe 2*: Do you usually get distracted by thoughts of the event? |
| 2. In your opinion, what do you think can be done to improve on security and healthcare access? |
| **C. Professional Perseverance of health workers during conflict** |
| 1. Have you thought of abandoning your post of duty? |
| *Probe 1*: What will make you stay back if you are planning to leave? |
| 2. With all that you have told me today, what are your motivations to keep carrying on daily in your profession? |
| *Probe 1*: Are there any added incentives at work? |
| *Probe 2*: How do you manage your work and family (If family)? |
| **D. Final thoughts** |
| We have come to the end of the interview, thank you for your participation. Is there anything else you would like us to know? |

A smart phone (Motorola g(8) plus 4104) was used to record interviews. After each session, the recording was saved and transferred to a personal computer (PC) where transcription and translation of texts were done remotely. Data anonymity was respected all through the data collection process.

## Data analyses

Interviews were transcribed using an inductive, iterative process involving: 1) familiarization with participants, questionnaire, and study area; 2) coding data by abstracting texts; 3) identifying concepts and themes based on research objectives 4) identifying recurring themes 5) refining and 6) report writing.

Transcription and translation of data were done manually using Microsoft word and reviewed by other members of the research team to avoid misinterpretation of texts. The transcribed data were then entered into Atlas.ti 9 software for coding texts segments. Codes were later used as themes and a total of 86 themes emerged which were later grouped according to pre-determined concepts (Table 3). Recurring themes were highlighted and used in the final analysis. Discrepancies in coding were examined by a fieldwork assistant and further reviewed by the fieldwork mentor where needed.

## Results

### Socio-demographic characteristics of participants

Participants' ages ranged from 27–40 with a mean of 34 years. Male and female genders were equally represented. Majority of respondents were nurses (Table 1).

**Table 3. Code–participant frequency.**

| Concepts/Topic | Respondent 1 QC = 33 | Respondent 2 QC = 41 | FG discussions QC = 56 | Total |
|---|---|---|---|---|
| **Barriers to healthcare** *QC = 59; **CC = 35 | 18 | 19 | 22 | **59** |
| **Consequences of limited healthcare** QC = 14; CC = 11 | 2 | 11 | 1 | **14** |
| **Experience with brutality** QC = 28; CC = 15 | 8 | 8 | 12 | **28** |
| **Motivation to work** QC = 13; CC = 10 | 6 | 3 | 4 | **13** |
| **Response to crisis** QC = 12; CC = 9 | 5 | 5 | 2 | **12** |
| **Totals** | **39** | **46** | **41** | **126** |

*QC: Quotation Count

**CC: Code (theme) Count.

The table below shows what participants said (codes/themes) about a particular concept (topic) and the number of times participants spoke about each theme (quotations). Half of what participants said pertained to barriers to healthcare utilization (47%) (Table 3).

## Experiences of health workers with brutality

When asked about 'experiences with brutality' from any of the armed groups, majority of participants reported that their lives had been threatened in one way or another; either at home, on their way to work, at their work places or during outreach activities (Table 4). Some participants said they had been assaulted in the process and one participant described an event involving the murder of two HCWs at home (Table 4). Two participants admitted being abducted and released only after a ransom payment. The threats, intimidations, assaults, and murders resulted in an overall insecurity in and around the hospital premises. This caused panic, leading to unplanned internal displacement of HCWs and their families to other regions for better opportunities.

## Barriers to healthcare delivery

When asked about the barriers to healthcare delivery, themes related to workers' experiences with brutality were repeatedly raised by participants, as it contributed to the global unrest in the community and at their work places (Fig 1). As a result of the insecurity, panic and uncertainty, health care workers frequently came late to work or did not come at all, most often for several days (Table 5). Others were overburdened with multiple shifts to cover for the absences

**Table 4. Responses and quotes to the question "have you had any physical encounter with an armed group?".**

| Document | Relevant quote | Codes |
|---|---|---|
| Respondent 2 | *'Another instance I was called that one staff was missing, she had to run away because she was receiving threats.'* | **Personnel threatened** |
| Respondent 1 | *'Yes, yes. Unfortunately one of our doctors was attacked although not in the health facility. He travelled to Bamenda to run an errand for the health facility where he was attacked and the fingers cut off. . . . . . . . .'* | **Physical assault of health workers** |
| FGD 1 | *'Yes, actually there is an incident we heard of in Mbengwi involving two nurses, a man and his wife shot by the armed forces'* | **Health workers murdered** |

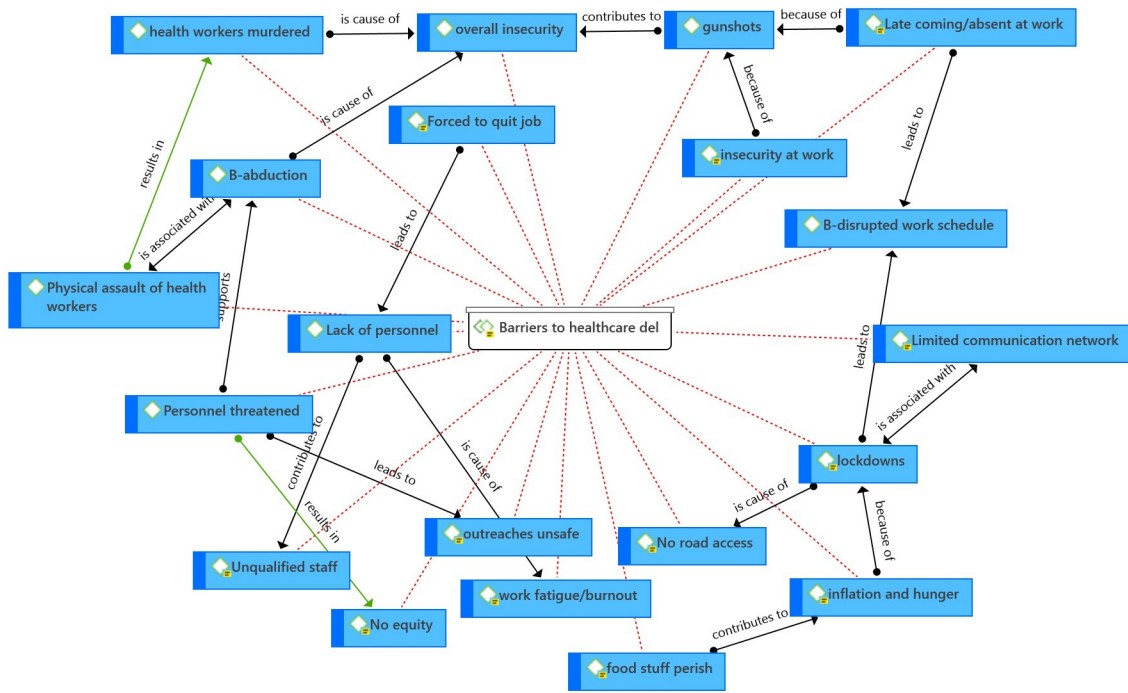

**Fig 1. Network of barriers to healthcare delivery.**

which led to physical and mental fatigue. The already existing problem of personnel shortage made the situation worse. In extreme cases, three participants expressed their resolve to leave and some fled eventually (Table 5). Recruiting competent health personnel became challenging and this contributed to the hiring of unqualified staff (Table 5).

Common problems like lack of communication network, internet services, incessant power cuts and perpetual lockdowns disrupted economic and hospital activities. Some participants sadly explained that the lockdowns led to inflation of prices and as such worsened the rate of hunger and lack of concentration amongst them during working hours (Fig 1).

Two participants painfully described how they had lost siblings amid the crisis, thus concentrating at work often became difficult. In the same light, focus at work was further impaired by gunned men within the hospital premises, even at bedside during administration of care to patients. Sometimes HCWs were unable to treat all patients equally because they were worried they will be met with resistance or even threats from another armed group for treating an enemy (Fig 1).

**Table 5. Responses and sample quotes to the question "how has this incident(s) affected the way you work?".**

| Document | Relevant Quote | Codes |
|---|---|---|
| Respondent 1 | 'Firstly is the insecurity. Most patients will not come to the hospital at a time when the atmosphere is tense and others will not like to visit this health centers especially with people (armed forces) patrolling the streets' | **Overall insecurity** |
| FGD 2 | 'There are also some patients who leave from far distances to reach the hospital and because of the insecurity; it is difficult for them to move' | **Overall insecurity** |
| Respondent 2 | 'Sometimes when the insecurity becomes so high, workers drop out and you are stuck with no one to work with. It becomes very difficult to recruit. . ...' | **Forced to quit job/ Lack of personnel** |

**Table 6. Probing into the question "do you think the incident has affected the way you work?".**

| Document | Relevant Quote | Codes |
|---|---|---|
| Respondent 2 | '. . .they escape into the bushes with the mosquitoes there. So they are exposed to bites, poor feeding like just cocoyam and palm oil, children become malnourished, they contract malaria and after the lockdown, the cases start coming in multitudes'. | **Disease complications Increase disease prevalence** |
| FGD 1 | '. . .most people are not able to reach the hospital, most women deliver at home. It is after weeks or a month of lockdown when the roads are accessible we start seeing critical cases. . ..' | **Delivery at home/disease complications** |

## Consequences of limited healthcare access

Participants shared the following thoughts that resulted from limited healthcare access during the ongoing conflict. They were quite disgruntled when talking about this subject.

Firstly, most participants thought that because of diminished healthcare utilization, there was an increase in the prevalence of chronic diseases such as hypertension, poorly controlled diabetes and post-traumatic stress disorder (PTSD) (Table 6). One participant said the cases of malnutrition in children under five and infectious diseases like malaria especially for those exposed to mosquito bites in the bushes rose significantly (Table 6).

Participants also discussed delayed diagnosis as one of the factors associated with limited healthcare access. As a result, patients are forced to take herbal remedies at home anticipating the day roads to the hospital facility will be accessible. They also pointed out that pregnant women at term had home births without proper postnatal care (Table 6). All these, they articulated, gave rise to complications that significantly increased maternal and infant mortality.

At the administrative level, a participant voiced the implications of a low patient attendance on hospital revenue and consequently medication availability. The issue of unplanned spending in order to cover for the weeks of lockdown was also raised.

## Health worker response to crisis

When asked about HCW response to the crisis, participants were all in favor of the crisis coming to an end so that life could return to normal. They felt a cease fire was the best way to end the conflict with the government being the main protagonist in ensuring this (Table 7). They also mentioned that the hospital had been a safe haven for health workers and the community at large, providing shelter and food with the help of some NGO's (Table 7).

One participant emphasized the importance of free aid provided by NGO's such as 'Doctors Without Borders' in the response, that they provide free food and clothing for the internally displaced and this makes rationing for all possible (Table 7). Some participants strongly

**Table 7. Responses and sample quotes to the question "what do you think can be done to improve on security and healthcare access?".**

| Document | Relevant Quote | Codes |
|---|---|---|
| Respondent 1 | 'There should be a cease fire between both parties then we are going to have peace. If that is done there will be peace and. . .' | **Ceasefire** |
| Respondent 2 | 'This is a place of refuge for many so during that time the health center is crowded with people and all the staff stays here until the period is over. So these are some of the ways they support and encourage us and it is good.' | **Safe haven** |
| Respondent 2 | 'There were so many children in the compound and thanks to some organizations who helped to buy food for the children' | **Free aid** |
| FGD 1 | 'So maybe if community health workers are placed at the disposal of every community they will be able to reach them and administer baseline medications on time' | **CHW** |

**Table 8.  Responses and sample quotes to the question "have you thought of abandoning your post of duty?".**

| Document | Relevant quote | Codes |
|---|---|---|
| FGD 1 | 'What is still keeping me here is my love for the job. I love my job; I love to care for the patients……' | **Love and passion** |
| Respondent 1 | Firstly if things normalize, if there is a cease fire, I will not leave and more so I would have gone if not for the encouragement given to us by the administrator. | **Motivation to work** |

perceived 'waiting patiently' was the best option since the anticipated intervention of the government had tarried.

Another participant explained that community health workers (CHW) be trained and sent to remote villages where they can quickly identify those critically ill and administer first aid or if possible transport the patient to the hospital) (Table 7). In her opinion that was the best way to curb disease complications and mortality (home deaths and infant mortality.

## Motivation to work

Participants gave different responses when asked about the factors that motivate health personnel to work. Most of the participants expressed their love and passion for the profession as a drive that makes them empathetic towards patients and derive satisfaction in their wellness (Table 8). One participant was determined to stay at all costs because she believed it was divinely ordained. Some however, insisted that their basic needs had to be met for them to keep working and according to a few the hospital administration played a major role in prolonging their stay (Table 8).

Four participants felt that being a health worker had a protective effect against brutality. Nevertheless, this contradicted some of the experiences shared concerning assaults on HCWs.

## Discussion

Armed conflicts are a major contributor to global disease burden (GDB) due to their direct effects via deaths and injury and their indirect effects on the educational and healthcare systems [9]. The objective of this study was to understand the experiences of HCWs with armed groups, the limitations to healthcare delivery during an armed conflict from HCW perspective and their response to these challenges. We provide an in-depth discussion of the thoughts and experiences of twelve HCWs from the North West region of Cameroon categorized into five major concepts; experiences with brutality, barriers to healthcare delivery, consequences of limited healthcare access, response to the crisis and motivation to work.

Violence in all its forms was the main theme that emerged from the experiences of HCWs with armed groups. Although our study did not suggest HCWs being specifically targeted by armed groups, almost all participants had one or more brutal experiences that suggested otherwise. A study on the effects of armed conflicts on population health also supported the fact that HCWs were specifically targeted by armed groups [10]. According to a systematic review of articles concerning HCWs in conflict settings, violence against HCWs was the most tackled theme [11]. An observational study in Cameroon estimated a total of 11 HCWs killed since the onset of the conflict [12]. The International Humanitarian Law (IHL) clearly states the conduct of parties to armed conflict which involves not targeting those providing medical or humanitarian assistance [13]. Our findings were in complete violation of the IHL rules.

Armed conflict diminishes healthcare utilization significantly evidenced by observational studies in Cameroon on vaccination coverage and ANC attendance which dropped considerably from 2017 to 2018 [12,14]. In our study, the topic 'barriers to healthcare delivery' had the

most emerging themes and participants mentioned the overall insecurity in the region as a major hindrance to healthcare delivery. In Afghanistan, almost 3/4[th] of patients had a limiting factor to healthcare, of which the main barriers cited were insecurity (60%), long distance and lack of transport [15]. Other themes mentioned by participants included shortage of personnel, HCWs overburdened with work, HCW fleeing to other regions and intimidation at work by armed groups. Similar challenges were documented in neighbouring Northeastern Nigeria by the International Peace Institute [5,16]. Further studies in ten underdeveloped conflict settings revealed that shortage in health workforce, health service delivery and insecurity are major barriers to implementing sexual, reproductive, child and adolescent interventions [17].

Limited healthcare access can have dire consequences especially on maternal and child health [18]. In our study, participants highlighted the implications of the former on women and children, most especially the acute complications such as malaria and malnutrition which are a common cause of death among children in conflict settings [18]. Between 1995 and 2015 it was estimated that more that 10 million children died due to the direct or indirect effects of conflict, with malnutrition and infections being among the most common causes [18,19]. Vaccination coverage and ANC attendance dropped by 35% (DPT3) and 18% respectively in the South West region of Cameroon within one year [12,14]. Although participants in our study and related studies emphasized on maternal and child health, it was also noticed that the health of adults deteriorated. Managing chronic diseases like diabetes and hypertension became challenging and there was an apparent surge in new cases as articulated by participants. Congruently, 15 patients with diabetes and hypertension in conflict Iraq reported consistent impedance to NCD care including shortages of medications and insecurity [20].

Armed conflict does not promote good health thus its cessation is unarguably the ultimate step in restoring the healthcare system to normal. States, governments, peace institutes, human rights and NGO's all play a part in curbing the damages and losses caused by armed conflicts [13]. They help alleviate suffering, save lives and provide humanitarian aid. As expected, participants consensually opted for a ceasefire as a means to end the conflict. The effects of armed conflict on health outcomes are better evaluated after the conflict has ended. Post-conflict Nepal made progress in 16 out of 19 Millennium Development Goals (MDG) health indicators [21]. This supports the ceasefire concept as possibly the best solution. The impact of NGOs was greatly felt according to participants' testimonies. Core humanitarian principles governing HCWs include neutrality, impartiality [13] and the obligation to ensure all the sick and wounded receive medical care. Nevertheless, HCWs in our study were intimidated into discriminative care and some were coerced to take sides with either of the armed groups. This reflects the real challenges HCWs face in implementing these principles on the field, which may explain why they have to either flee or succumb to threats from armed groups.

Behavioral studies have identified factors that promote job satisfaction such as conducive work environment, career progression or promotion, remuneration, adequate working conditions and recognition [22,23]. In conflict setting however, motivations may change such as is the case in our study, where participants' main motivation was their moral obligation to improve lives and reduce suffering and their ability to lead a relatively comfortable life. One participant emphatically expressed her religious inclinations towards patients and the community. Similar coping strategies were identified amongst HCWs in Yemen including fatalism [24] and in Syria participants had both intrinsic (humanitarian) and ideological (religious) reasons for staying at work [23]. Hospital administration has a vital role in ensuring job satisfaction amongst workers and according to our study the hospital administration did not fail in this aspect.

## Conclusion

Health care workers face significant personal and professional challenges in conflict settings as evidenced in our study, despite existing regulations governing conduct during conflicts and wars. This affects their performance and consequently healthcare delivery; as such they desperately find opportunities to flee the hostile environment which is usually elusive. They are therefore forced to exercise perseverance to survive, even to their own detriment. There is need for an independent body to enforce International Health regulations and attend to the physical and psychological needs of HCWs in conflict zones.

## Supporting information

**S1 Table. Socio-demographic characteristics of participants (SPSS version 20).**
(DOC)

**S2 Table. Quotation report–assault of HCWs.**
(PDF)

**S3 Table. Quotation report–HCWs murdered.**
(PDF)

**S4 Table. Quotation report–overall insecurity.**
(PDF)

**S5 Table. Quotation report–disease complications.**
(PDF)

**S6 Table. Quotation report–delivery at home/malnutrition.**
(PDF)

**S7 Table. Quotation report–safe haven.**
(PDF)

**S8 Table. Quotation report–free humanitarian aid.**
(PDF)

**S9 Table. Quotation report–cease fire.**
(PDF)

**S10 Table. Quotation report–Community Health Workers (CHW).**
(PDF)

**S11 Table. Quotation report–motivation to work.**
(PDF)

## Acknowledgments

We would like to appreciate the entire staff of the Saint Joseph Catholic Hospital Widikum as well as the District Hospital Batibo for their collaboration to make this study a reality. We would also like to thank the "Medical Research and Career" research team who contributed to make this work a success.

## Author Contributions

**Conceptualization:** Juste Ongeh Niba, Stewart Ndutard Ngasa.

**Data curation:** Juste Ongeh Niba, Anne-Marie Awa.

**Formal analysis:** Juste Ongeh Niba.

**Funding acquisition:** Juste Ongeh Niba, Carlson Babila Sama.

**Investigation:** Juste Ongeh Niba.

**Methodology:** Juste Ongeh Niba, Stewart Ndutard Ngasa.

**Project administration:** Juste Ongeh Niba, Stewart Ndutard Ngasa, Carlson Babila Sama.

**Resources:** Juste Ongeh Niba, Anne-Marie Awa.

**Software:** Juste Ongeh Niba.

**Supervision:** Stewart Ndutard Ngasa, Therence Nwana Dingana.

**Validation:** Stewart Ndutard Ngasa, Therence Nwana Dingana.

**Visualization:** Juste Ongeh Niba, Neh Chang.

**Writing – original draft:** Juste Ongeh Niba.

**Writing – review & editing:** Juste Ongeh Niba, Stewart Ndutard Ngasa, Neh Chang, Eric Sanji, Therence Nwana Dingana, Leticia Tchouda, Mekolle Enongene Julius.

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
