## [Decision Letter · Decision Letter 0]

30 Jul 2022

PGPH-D-22-01049

Conflict, Healthcare and Professional Perseverance: A Qualitative Study in an Anglophone Region in Cameroon

Dear Dr. Niba,

Thank you for submitting your manuscript to PLOS Global Public Health. After careful consideration, we feel that it has merit but does not fully meet PLOS Global Public Health’s publication criteria as it currently stands. Therefore, we invite you to submit a revised version of the manuscript that addresses the points raised during the review process.

We look forward to receiving your revised manuscript.

Kind regards,

Shailendra Prasad, MD, MPH

Academic Editor

Journal Requirements:

1. Please ensure that you refer to the main tables in your text as, if accepted, production will need these reference to link the reader to the tables.

Additional Editor Comments (if provided):

Dear Authors,

I would encourage you to revise and resubmit. Please address the concerns that the reviewers have on your paper. Looking forward to reading the resubmission.

Reviewers' comments:

Reviewer's Responses to Questions

**Comments to the Author**

1. Does this manuscript meet PLOS Global Public Health’s publication criteria? Is the manuscript technically sound, and do the data support the conclusions? The manuscript must describe methodologically and ethically rigorous research with conclusions that are appropriately drawn based on the data presented.

Reviewer #1: Partly

Reviewer #2: No

2. Has the statistical analysis been performed appropriately and rigorously?

Reviewer #1: Yes

Reviewer #2: No

3. Have the authors made all data underlying the findings in their manuscript fully available (please refer to the Data Availability Statement at the start of the manuscript PDF file)?

Reviewer #1: Yes

Reviewer #2: Yes

4. Is the manuscript presented in an intelligible fashion and written in standard English?

Reviewer #1: Yes

Reviewer #2: Yes

5. Review Comments to the Author

Reviewer #1: Great Job to the team for the progress made so far!

Just a few comments, observations and suggestions for your consideration.

1, It would have been good to tie this study to its relevance in the public health spaces in terms of real life existing problems it seeks to solve or gaps to be addressed. e.g " to understand the experiences of 12 HCWs in the North West region of Cameroon with armed groups; evaluate how it affects healthcare delivery from HCWs perspective and examine HCWs coping mechanisms during the conflict with a view to informing health worker protection policies in conflict zones". You alluded to the need to better enforce the IHR implementation in your conclusion, which is good. I think this also needs to be reflected in the abstract and Introduction sections

2,The Sample size could have been better more. 12 HCW in one district and mostly from one facility seems to me not very representative for the conclusions drawn.

3, The survey tool could have been richer if there was a segment to sought to understand specific changes or actions respondents would have desired. This is not a major limitation however.

Reviewer #2: The research is technically weak, lacks detail in several areas, and the rigor required for qualitative research.

Introduction – the authors must provide a bit more detail of what the Anglophone crisis in Cameroon is. Readers must get all the details about the crisis from this article without resorting to an internet search to find out what this crisis is about.

Setting – It is not clear whether “Batibo Health District” is a health facility (hospital), or an administrative district with multiple health facilities. While this becomes somewhat clear that the health district may mean a district hospital, this must be clarified since “Health District” can have many meanings depending on country.

Participants – The study details how participants were recruited from the “district” based on a criterion yet in the results, it becomes somewhat apparent that these healthcare workers all worked at one rural hospital. The authors should provide a breakdown (percentages) of the various categories of healthcare workers at the hospital (nurses, doctors, lab scientists) as these were the only ones recruited for the study, to justify the claim that indeed care was taken to have each group fairly represented.

Data collection – this section is lacking in detail. It is not clear how many sessions were conducted for the FDGs. The authors attempt to explain data collection for both FDGs and interviews in the same sentences, leaving the reader unclear whether the entire data collection for the study was done in just over an hour.

6. PLOS authors have the option to publish the peer review history of their article (what does this mean?). If published, this will include your full peer review and any attached files.

**Do you want your identity to be public for this peer review?** For information about this choice, including consent withdrawal, please see our Privacy Policy.

Reviewer #1: **Yes: **Dr Nonso Ephraim Ejiofor. DDS, MBA.

Reviewer #2: No

---

## [Editor Report · Decision Letter 1]

17 Oct 2022

PGPH-D-22-01049R1

Conflict, Healthcare and Professional Perseverance: A Qualitative Study in a Remote Hospital in an Anglophone Region of Cameroon

Dear Dr. Niba,

Thank you for submitting your manuscript to PLOS Global Public Health. 

Please pardon my "minor revision" recommendation. You have addressed the concerns that the reviewers had. In some instances these have been addressed a little too well!

My recommendation would be to cut some of the changes that you have made and still address the reviewers comments -

1) The second paragraph of the Introduction: I recommend you delete "Over the years, it appeared that the autonomy and identity of the Southern Cameroonians (today’s NW and SW regions) was gradually being suppressed in an attempt by their French counterparts to have absolute control. Awareness of this stirred grievances and complaints which were apparently not well handled by the current regime." and "eventually a problem which seemed so small "

2) The fourth paragraph of the introduction: I recommend you move the sentence "The ultimate goal is to provide local authorities and stake holders with comprehensive evidence that will allow them make informed decisions concerning the welfare of HCWs in conflict settings." to after the three questions. I think the sentence is important; it reads better after the three questions.

We look forward to receiving your revised manuscript.

Kind regards,

Shailendra Prasad, MD, MPH

Academic Editor
---

## [Editor Report · Decision Letter 2]

21 Oct 2022

Conflict, Healthcare and Professional Perseverance: A Qualitative Study in a Remote Hospital in an Anglophone Region of Cameroon

PGPH-D-22-01049R2

Dear Dr Niba,

We are pleased to inform you that your manuscript 'Conflict, Healthcare and Professional Perseverance: A Qualitative Study in a Remote Hospital in an Anglophone Region of Cameroon' has been provisionally accepted for publication in PLOS Global Public Health. Thank you for addressing the suggestions and comments promptly. 

Best regards,

Shailendra Prasad, MD, MPH

Academic Editor
